# Congenital SARS-CoV-2 Infection in Two Neonates with Confirmation by Viral Culture of the Placenta in One Case

**DOI:** 10.3390/v15061310

**Published:** 2023-05-31

**Authors:** Joseph V. Vayalumkal, Amuchou S. Soraisham, Ayman Abou Mehrem, Anirban Ghosh, Jessica K. E. Dunn, Kevin Fonseca, Hong Zhou, Byron M. Berenger, Elaine S. Chan, Marie-Anne Brundler, Yi-Chan Lin, David H. Evans, Sharon Rousso, Verena Kuret, John M. Conly

**Affiliations:** 1Department of Pediatrics, Alberta Health Services, University of Calgary, Calgary, AB T2N 4N1, Canada; 2Alberta Children’s Hospital Research Institute, University of Calgary, Calgary, AB T2N 4N1, Canada; 3Alberta Public Health Laboratory, Alberta Precision Laboratories, Calgary, AB T2N 4W4, Canada; 4Department of Microbiology, Immunology & Infectious Diseases, University of Calgary, Calgary, AB T2N 4N1, Canada; 5Department of Pathology and Laboratory Medicine, University of Calgary, Calgary, AB T2N 4N1, Canada; 6Department of Medical Microbiology and Immunology, University of Alberta, Edmonton, AB T6G 2E1, Canada; 7Department of Pediatrics, Red Deer Regional Hospital, Red Deer, AB T4N 4E7, Canada; 8Department of Obstetrics and Gynecology, University of Calgary, Calgary, AB T2N 4N1, Canada; 9Department of Medicine, Alberta Health Services and University of Calgary, Calgary, AB T2N 4N1, Canada; 10Snyder Institute for Chronic Diseases and O’Brien Institute for Public Health, University of Calgary, Calgary, AB T2N 4N1, Canada

**Keywords:** congenital, transplacental, SARS-CoV-2, neonatal, pregnancy, viral culture, COVID-19

## Abstract

Congenital infections with SARS-CoV-2 are uncommon. We describe two confirmed congenital SARS-CoV-2 infections using descriptive, epidemiologic and standard laboratory methods and in one case, viral culture. Clinical data were obtained from health records. Nasopharyngeal (NP) specimens, cord blood and placentas when available were tested by reverse transcriptase real-time PCR (RT-PCR). Electron microscopy and histopathological examination with immunostaining for SARS-CoV-2 was conducted on the placentas. For Case 1, placenta, umbilical cord, and cord blood were cultured for SARS-CoV-2 on Vero cells. This neonate was born at 30 weeks, 2 days gestation by vaginal delivery. RT-PCR tests were positive for SARS-CoV-2 from NP swabs and cord blood; NP swab from the mother and placental tissue were positive for SARS-CoV-2. Placental tissue yielded viral plaques with typical morphology for SARS-CoV-2 at 2.8 × 10^2^ pfu/mL confirmed by anti-spike protein immunostaining. Placental examination revealed chronic histiocytic intervillositis with trophoblast necrosis and perivillous fibrin deposition in a subchorionic distribution. Case 2 was born at 36 weeks, 4 days gestation. RT-PCR tests from the mother and infant were all positive for SARS-CoV-2, but placental pathology was normal. Case 1 may be the first described congenital case with SARS-CoV-2 cultivated directly from placental tissue.

## 1. Introduction

The novel coronavirus disease 2019 (COVID-19) caused by the Severe Acute Respiratory Syndrome Coronavirus 2 (SARS-CoV-2) emerged in December of 2019 in Hubei province, China and rapidly disseminated across the globe [1]. Like most other respiratory infections, large droplets, fomites, and aerosols were thought to be the primary contributors to transmission. It had originally been postulated that vertical transmission was not a significant mode of transmission for COVID-19 [2,3,4,5]. Early reports from China were conflicting but raised the possibility of in utero infection through transmission via the placenta [3,6,7]. With time, further evidence emerged supporting transmission to neonates of mothers with active SARS-CoV-2 infection [8,9,10,11,12,13,14]. Vertical transmission has been previously demonstrated by PCR positive newborns within 48 h of birth, immunohistochemistry, and visualization of viral particles in the placenta by electron microscopy (EM). No reports to date have cultured virus from the placenta. We present two cases that demonstrate transplacental transmission of SARS-CoV-2, one of which had positive viral cultures for SARS-CoV-2 from the placenta.

## 2. Materials and Methods

Health records were reviewed for details of clinical presentation and management. Standardized methods were used for histopathology [15], molecular diagnostics [16] and viral cultures [17]. Nasopharyngeal (NP) swabs in universal/viral transport media and clinical samples were collected from the mother and baby. The umbilical cord and placenta were stored in a refrigerator at 4 °C for approximately five days until it was moved to a −80 °C freezer. The sample was then retrieved two months later, thawed and portions of the cord and placenta were split and transported to the Level 3 Laboratory on ice prior to culture and further testing. Placentas were sent for pathological examination as per provincial guidelines for diagnosis and management of maternal COVID infection. Immunohistochemical staining was conducted on formalin-fixed, paraffin-embedded 4-μm sections following Tris/EDTA pH 9.0 antigen retrieval and peroxidase block. Immunohistochemistry for SARS-CoV-2 was conducted using a rabbit polyclonal antibody to the SARS-CoV-2 spike protein (1:50; Sino Biological, catalog number 40150-T62-COV2) on the BOND III (Leica Biosystems; Buffalo Grove, Illinois). Electron microscopy was also performed on placental tissue samples [18].

Reverse transcriptase real-time PCR (RT-PCR) testing and whole-genome sequencing (WGS) were performed on clinical samples to determine the Pangolin lineage and relatedness.

The Simplexa Direct COVID-19 RT-PCR (Diasorin, Molecular) was performed per manufacturer’s instructions. The E gene RT-PCR was performed per locally developed and validated assay by the Provincial Laboratory for Public Health (ProvLab) [16]. Samples with a cycle-threshold (Ct) value < 35 were considered positive. If the Ct was ≥35, amplification from the same eluate was repeated in duplicate and was considered positive if at least two of the three results had a Ct < 41. The N gene US CDC RT-PCR was performed as previously described [17].

### 2.1. Viral Cultures

Viral cultures were performed (Case 1) on NP swabs, placental tissue, umbilical cord tissue and cord blood using standard methods [19]. In the Level 3 Laboratory, samples of tissue homogenate were diluted 10-fold and used to infect Vero cells. Three days post-infection of the culture, the cell-culture plates were fixed and stained with a crystal violet/formaldehyde solution to visualize the viral plaques. To confirm the identity of the virus seen, the plates were destained with ethanol, immunostained with a 1:500 diluted rabbit anti-SARS-CoV-2 spike antibody (Prosci Inc., cat#3525) and visualized with a secondary goat anti-rabbit IgG (H+L) antibody coupled to horseradish peroxidase (Invitrogen, cat#G21234) and KPL TrueBlue peroxidase substrate (SeraCare, cat#5510-0052).

### 2.2. Consent and Ethics

Consent was obtained for additional testing of clinically collected samples through a Research Ethics Board CHREB #20-0444 approved protocol from the University of Calgary; SARS-CoV-2 plaque assays were performed through a Research Ethics Board approved protocol (#Pro00099761) from the University of Alberta. Consent was also obtained from the mothers to publish a case report.

## 3. Results

### 3.1. Case Presentation

#### 3.1.1. Case 1 (December 2020)

The mother was 35 years of age with no significant past medical history and one previous pregnancy resulting in a live birth. Antenatal ultrasounds showed a singleton pregnancy with no fetal anomalies. There was spontaneous rupture of membranes at 28 weeks and 5 days gestation and oligohydramnios was noted. The mother received erythromycin for preterm pre-labour rupture of membranes (pPROM) as well as a single course of betamethasone. She presented to a community hospital at 30 weeks’ gestation with fever (38.6 °C) and a sore throat. An NP swab was collected for COVID-19 testing.

As there was clinical suspicion of chorioamnionitis, she also received intravenous (IV) cefazolin and metronidazole. She had a hemoglobin of 111 g/L, a platelet count of 177 × 10^9^/L and a lymphocyte count of 0.7 × 10^9^/L. She was transferred to a tertiary maternal–fetal medicine centre, based on the prematurity, abnormal fetal heart rate Category II [20], (See Appendix A Appendix A) pPROM and pending COVID-19 status. Once the COVID-19 NP RT-PCR result was found to be positive (See Table 1), the antibiotics were discontinued. The maternal infection was mild and did not require COVID-specific therapy.

Spontaneous vaginal delivery was uncomplicated, with the neonatal resuscitation team attending with appropriate personal protective equipment (PPE). A male infant was born at 30 weeks 2 days gestational age; there was delayed cord clamping for 60 s. The Apgar scores were 7 at one minute and 9 at five minutes. Arterial cord blood pH was 7.26. His birthweight was 1.83 kg (91st percentile) and the head circumference was 28.5 cm (69th percentile). The baby did not have any contact with the mother in the immediate post-natal period. He was brought straight to the resuscitation warmer and started on continuous positive airway pressure (CPAP) at 2 min of life. He was then transferred to the neonatal intensive care unit (NICU) for ongoing care.

Based on the prematurity and pPROM, a partial sepsis workup was initiated, and the neonate was started on ampicillin and gentamicin. He was stable for the first two days of life, weaned to room air and gradually increased to full feeds. The maximum supplemental oxygen requirement was 40%. The chest radiograph showed lung volumes mildly diminished, but the lung and pleural spaces were clear. The baby had NP swabs collected at 24 and 48 h of life that tested positive for SARS-CoV-2 (Table 1). Blood cultures yielded no growth and antibiotics were discontinued. See Appendix A Appendix A for the results of other laboratory investigations.

From a respiratory standpoint, the baby was weaned off CPAP to room air by day two of life and then remained in room air until a clinical deterioration on day five, with hypothermia and signs of respiratory distress. An irregular respiratory pattern was noted, and there were signs of increasing work for breathing. Blended oxygen therapy was started at 1 L/min of high flow via a nasal cannula. The baby appeared lethargic and large milky aspirates were noted from the nasogastric suction. A workup for sepsis was initiated. A lumbar puncture was performed but it was contaminated with blood (See Appendix A Appendix A for the results of investigations). SARS-CoV-2 RT-PCR on the CSF sample was negative (Table 1). The baby was treated for suspected sepsis with ampicillin and cefotaxime for 10 days and seemed to improve with antibiotic treatment. The mother continued to express her breast milk and it was provided to the baby as feeds were progressing. The parents were regularly updated via virtual ward rounds and the mother was permitted to visit the NICU after day 11 of life when she was considered asymptomatic and confirmed to be negative for SARS-CoV-2 by NP swab.

The SARS-CoV-2 RT-PCR NP swabs repeated on Day 14 and Day 21 from the baby were still positive, although increasing cycle-threshold (Ct) values suggested a reduction in the viral load.

By Day 28 of life, the baby was transferred to the level II neonatal intensive care at a community hospital in room air. Another NP swab for SARS-CoV-2 detection at day 32 was negative. He was discharged home on day 54 (corrected 38 weeks) with a weight of 3.12 kg (47th percentile).

#### 3.1.2. Case 2 (January 2021)

The mother was 33 years of age with symptomatic COVID-19 infection. This was her third pregnancy, and it was uncomplicated; there was no exposure to alcohol, drugs or tobacco products. The mother did not have gestational hypertension or diabetes. Antenatal ultrasounds were normal. An ultrasound at 34 weeks was done due to measuring small for dates, but growth was noted to be at the 75th percentile. The mother was taking prenatal vitamins, magnesium, and iron throughout the pregnancy.

She presented to the hospital with worsening symptoms on day eight of the illness. She had fever and there was no rupture of membranes, no uterine tenderness nor foul-smelling discharge. Her CBC showed a CRP of 13 mg/L, hemoglobin of 119 g/L, platelets of 116 × 10^9^/L, and normal ALT, AST and APTT. The fetal status was non-reassuring, with fetal tachycardia noted on the nonstress test (NST), repeated decelerations, and decreased variability (Category III) [20] (See Appendix A Appendix A). Due to the persistent fetal tachycardia, an emergency caesarean section was performed. Throughout her course the mother had very mild symptoms, did not require respiratory support, nor treatment specific for COVID-19 infection and did not require admission to the intensive care unit. She was discharged from the hospital two days after her caesarean section.

A male infant was delivered by emergency caesarean section at the gestational age of 36 weeks and 4 days. The neonatal resuscitation team attending the delivery used appropriate PPE. There was no contact between the mother and neonate in the immediate period after the delivery. Apgar scores were 8, 8, and 9 at one, five, and ten minutes respectively. Arterial cord blood pH was 7.33. The birth weight was 3.330 kg (90th percentile) and head circumference 34.5 cm (84th percentile). There was no delayed cord clamping. The neonate was brought to the warmer, dried and stimulated. The neonate initially cried with good tone but appeared dusky. At five minutes of life, he was noted to have mild increased work for breathing and required CPAP support. Meconium was passed at 10 h of age. He was taken to the NICU, with a maximum oxygen requirement of 23% for the first 36 h of life. The clinical picture was that of transient tachypnea of the newborn. He did not develop apnea of prematurity and remained on room air from day two of admission for the rest of his hospital stay. An NP swab at 24 h of age was positive for SARS CoV-2 by RT-PCR (Table 2). The swab was repeated after 48 h of age and was positive again (Table 2).

His symptoms of COVID-19 included mild rhinorrhea, nasal congestion, foul smelling stools as well as lethargy for the first two weeks of life. He received only infant formula. His neurological examination was normal. A troponin level was checked on day of life nine along with CRP, which were both normal (See Appendix A Appendix A for results of other laboratory investigations). There were no other clinical concerns. He was discharged home at 20 days of life with no known sequelae.

### 3.2. Virology

The results of the PCR testing, WGS, and quantitative viral cultures of the specimens obtained from the various clinical samples obtained from Case 1 are provided in Table 1 and Table 2. The WGS revealed no single nucleotide variant (SNV) differences in the sequences from the mother, the infant, and the placenta in Case 1. Viral cultures and WGS were not performed on the specimens from Case 2 because they were not available to the Level 3 laboratory. Viral plaques were identified from the viral cultures (Figure 1 and Figure 2). These plaques were noted to have a halo-like appearance that is characteristic of SARS-CoV-2 cultured on Vero cells with a 1% carboxymethylcellulose overlay. Microscopy showed no bacterial or fungal contamination.

### 3.3. Histopathology and Electron Microscopy

Figure 3 and Figure 4 provide details on the histopathology and electron microscopy. In Case 1, the placenta showed a moderate infiltrate of histiocytes in the intervillous space (i.e., chronic histiocytic intervillositis), in a predominantly subchorionic distribution. The chronic histiocytic intervillositis is accompanied by trophoblast necrosis, mild chronic villitis, and perivillous fibrin deposition. The SARS-CoV-2 spike protein immunohistochemical stain was positive in a subset of trophoblasts, primarily in the subchorionic region (Figure 3). Electron microscopy revealed viral particles within the syncytiotrophoblasts (Figure 4). In Case 2, there was no evidence of inflammation in the placental tissue examined, including no intervillositis and/or chronic villitis noted.

## 4. Discussion

### 4.1. Principal Findings

These two cases represent confirmed transplacental transmission of SARS-CoV-2 with congenital infection. Both cases illustrate important observations that enhance our understanding of SARS-CoV-2 during pregnancy and in the neonatal period. Case 1 is quite informative given the combination of histopathology, electron microscopy, and WGS, with the addition of a viral culture as definitive evidence demonstrating the presence of SARS-CoV-2 in the placenta, which has not previously been described in the literature. Furthermore, the WGS confirmed that the lineage of the virus was identical for mother, placenta and neonate, providing incontrovertible evidence of vertical transmission, likely via the bloodstream. Case 2 highlights the importance of Ct values in different tissues and body fluids as well as the variation that can be seen with placental histopathology.

It is important to recognize the Ct values of both Case 1 and Case 2 from the placenta and cord blood, respectively, since they represent values that correlate with the cultivatable virus [17]. Ct values that are ≤25 have been cited as the likely threshold for cultivatable and, hence, transmissible virus [17]. As noted (Table 1 and Figure 2), for Case 1 the cord tissue and cord blood Ct values were over 25 and did not yield cultivatable virus, yet the placenta Ct values were lower than 25 and cultivatable SARS-CoV-2 was found. In comparison, Case 2 provides a unique and important finding whereby the placenta and cord Ct values were 24.9 and 19.5, respectively, inferring that if the viral culture were available, the growth of SARS-CoV-2 would be expected. Ct values from the cord blood or placenta could be an important tool to predict infectivity of the virus to the newborn. Further studies are required to determine if this observation holds true in other cases.

Both cases fulfil the proposed WHO definition of “Confirmed” in utero SARS-CoV-2 transmission, as published in February 2021 [21]. Previously, it was proposed that the diagnosis of early-onset neonatal SARS-CoV-2 infection should be limited to neonates with positive PCR tests in the initial 72 h of life as well as identification of SARS-CoV-2 in chorionic villus cells using immunohistochemistry or nucleic acid methods such as in situ hybridization [22]. Both Case 1 and Case 2 fulfill the first criteria, with the additional evidence from electron microscopy in Case 1. The images are consistent with electron microscopic findings of coronaviruses as described in a recent review [23].

Furthermore, our cases have shown marked differences in placental pathology. Case 1 revealed significant abnormalities in the placenta whereas Case 2 showed normal placental histopathology, suggesting compartmentalization of invasive infection of the placenta. Despite this difference, both neonates had congenital infections. This has been noted in other studies, including a study from India reporting normal placentas in just over 50% of the 179 cases examined [24]. The authors described further that only 2 of 15 that were tested for SARS-CoV-2 were positive for the virus and the morphologic examination of 1 of these placentas was normal [24].

Congenital infection is an uncommon occurrence but is now recognized with COVID-19. Multiple systematic reviews and meta-analyses have been conducted about pregnancy and neonatal outcomes. Papapanou et al. reported that when studies of moderate or high quality were considered, neonatal PCR positivity rates ranged from 2% to 7%, with the rate of 2.5% representing the largest sample [5]. Another systematic review which included 106 studies assessing vertical transmission of SARS-CoV-2, of which 66 were primary studies, concluded that vertical transmission is possible but not frequent [25]. The authors reported that of the included cohort and case series studies, 65/2391 (2.7%) neonates born to mothers with a diagnosis of COVID-19 tested positive for SARS-CoV-2 within 24 h of birth [25], which is consistent with the findings of Papapanou et al. [5].

The nature of placental abnormalities in congenital infection is also an area of ongoing investigation. In Case 1, there was additional evidence of placental abnormalities that supported in utero transmission of infection. However, in Case 2, placental abnormalities were not identified, either due to sampling error from “skip” lesions in the placenta or the absence of placental changes. Placental abnormalities are not necessarily predictive of infection in the neonate. In some studies, however, there have been more consistent abnormalities noted in the placental histopathology, such as chronic histiocytic intervillositis and syncytiotrophoblast necrosis, which may be risk factors for transplacental infection [24,26]. Other studies have suggested that the timing of infection may play a role in the patterns noted in placental pathology, where acute infections (<14 days) from delivery more frequently showed fetal vascular malperfusion lesions compared with non-acute infections [27].

Our study demonstrating different placental histopathology results matches observations from authors in the United States who reported on a series of 19 placentas from COVID-19-positive mothers where a spectrum of pathologies was noted [28]. Variability was also noted in a review of placental morphology and histopathological lesions associated with SARS-CoV-2 infections, where 42 articles were included [29].

The issue of peripartum transmission is often raised when neonates are found to be infected early on but given the placental abnormalities in Case 1 and the lack of rupture of the membranes in Case 2, this is improbable. Perinatal acquisition related to prolonged rupture of membranes is considered possible, but unlikely. Previous studies have shown variable results. In early studies from China, no virus was detected in vaginal fluids or the female genital tract of pregnant women [30,31]. In other reports, the virus was identified in a small percentage of vaginal swabs collected from pregnant women [32,33]. In our cases, post-natal acquisition via droplets or aerosols is also unlikely since the neonates did not have contact with the mother and were found to be positive within 24 h of birth.

### 4.2. Clinical Implications

Our report highlights several clinical issues. First, based on the low Ct values found in the two neonates described in our study, the viral load in the babies’ nasopharynx would be relatively high [17]. The findings of low Ct values early in the course of infection provide support to current infection control precautions for neonatal resuscitation and neonatal intensive care units when caring for neonates with suspected SARS-CoV-2 infection, which includes the use of appropriate personal protective equipment. Healthcare workers must be aware that even neonates could be contagious, given the quantity of virus identified from the nasopharynx.

Second, the diagnosis of placental infection with SARS-CoV-2 should include viral cultures as an option. A consensus document by Roberts et al. described a standardized definition of placental infection with SARS-CoV-2 [34]. Experts recommended that placental infection be defined using techniques that allow for virus detection and localization in placental tissue by several different methods [34]. Interestingly, viral culture was not listed among these methods. The World Health Organization scientific brief regarding the definition and categorization of the timing of mother-to-child transmission of SARS-CoV-2 also does not mention viral culture. This may be related to the resources required and the lack of access to viral cultures in most health care laboratory settingsViral culture from placental tissues of SARS-CoV-2 provides definitive evidence of placental infection and should also be included in such definitions.

Furthermore, the clinical presentation of neonates who have congenital infection with SARS-CoV-2 is highly variable. In Case 1, the baby had suspected sepsis symptoms which may have been related to SARS-CoV-2 infection. No other cause for the clinical deterioration was identified, but this is common among preterm neonates. In Case 2, the symptoms were quite mild and included symptoms of the upper respiratory tract which are common among older infants and children. Many neonates with suspected congenital infection may be asymptomatic [35,36,37]. In neonates with post-natal acquisition, a variety of symptoms have been described including apneas, poor feeding, lethargy, respiratory distress, vomiting and diarrhea [38]. This non-specific presentation is common to most viral and bacterial infections in neonates. Laboratory anomalies in neonates may include leukocytosis, lymphopenia, thrombocytopenia, and elevated inflammatory markers [39]. In a prospective UK population-based cohort study, 66 neonates were identified with confirmed SARS-CoV-2 infection [40]. Of these babies, 42% (16) had severe neonatal infection as defined by criteria published by Dong and colleagues [41] and 36% received care in NICU or PICU. The authors reported that two (3%) were considered to have possible vertically acquired infection (SARS-CoV-2-positive sample within 12 h of birth where the mother was also positive), based on published criteria [42]. Common signs at presentation that were noted in over 30% of neonates included: fever, poor feeding or vomiting, coryza, respiratory distress, lethargy, tachypnea, and supplemental oxygen requirement [40]. The neonates described in our cases certainly fit with this non-specific clinical presentation, although did not have the laboratory anomalies identified in other neonates.

### 4.3. Research Implications

Our cases highlight several important areas where further study is required. The clinical risk factors that lead to congenital infection are not yet well established. Further study is required to help understand which maternal SARS-CoV-2 infections predispose to in utero transmission and neonatal infection. Additional research is required to clarify the mechanism of placental infection and in-utero transmission. A study which included 31 pregnant women who delivered after 22 weeks and tested positive for SARS-CoV-2 during their pregnancy reported that only one case of placental infection was detected [43]. The authors suggest that trophoblasts are not likely to be infected by SARS-CoV-2 at term, but they do raise concern about preterm infection. Both of our cases were born preterm and this association of placental infection with viral replication in the syncytiotrophoblast appears to be noted in the preterm placenta, while the mature syncytiotrophoblast appears to be protected even though viral receptors such as ACE-2 are expressed. A combination of immunofluorescence staining, and electron transmission microscopy showed the presence of aggregates of viral particles in infected cells with a morphologic pattern closely similar to the placental tissue in Case 1 in our study, thus further emphasizing that the placenta was playing an active role in some congenital infections. A review on the structural and immunomodulatory defenses against SARS-CoV-2 in the human placenta, concluded that vertical transmission may occur but rarely, because of the potent physical barrier, and immunomodulatory mechanisms associated with the placenta [44]. It suggested the placenta may play a key role to diminish the immune response and reduce the cytokine storm associated with severe COVID-19 and potentially reduce SARS-CoV-2 transmission [44]. Certainly, in our cases, no severe presentations or outcomes were observed, and it is possible that placental defenses and timing of infection in relation to birth could play a role.

It is also of clinical interest that lower Ct values from the cord blood, cord tissue or placenta could be a marker of infectivity to the newborn. There is potential that low values could be used to predict congenital infection with SARS-CoV-2. This level of analysis with Ct values needs to be reproduced in future studies so that we can have a better understanding of viral transmission dynamics from mothers to babies.

The clinical presentation of neonates with congenital infection is also in need of further investigation. There may be many confounding factors including prematurity, and other non-infectious neonatal issues that make it difficult to understand the true clinical manifestations of SARS-CoV-2 infections.

Furthermore, the infectivity of congenitally infected neonates has not been studied. With high viral loads in the nasopharynx as in our study, transmission to close contacts is possible when appropriate personal protective equipment and infection prevention protocols are not followed.

### 4.4. Strengths and Limitations

Our report highlights the importance of viral cultures in assessing the potential for transplacental transmission. Molecular studies have limitations particularly when understanding viability and transmissibility of the virus in various body sites. Culturing the virus provides another level of reference when assessing the infectious potential of individuals, however this technique requires additional time and specialized resources and therefore is unavailable in most centers. In a systematic review assessing infectivity of SARS-CoV-2, the authors included 29 studies, all of which were case series [45]. Most of the studies analyzed respiratory samples. However, other samples were also included including fecal, blood and environmental samples. The placenta was not one of the tissues included in any of the studies in the meta-analysis. Our report highlights the infectious potential of placental tissue in the context of SARS-CoV-2 infection.

As noted previously, the recent consensus statement on placental infection does not mention viral cultures and previous publications did not culture virus [34]. Case 1 illustrates the utility of viral culture and WGSin confirming transplacental SARS-CoV-2 infection. Viral culture confirmed that the virus was viable and replicating in the sample tested.

Unfortunately, we were not able to perform viral culture on the placental sample from Case 2. The case was from a peripheral community hospital and the sample was not directed to the Level 3 laboratory for viral culture in a timely manner. It highlights the practical challenges of using viral culture routinely in most healthcare settings.

Furthermore, additional blood samples collected serially from mothers and neonates may have provided additional data on viral dynamics and infectivity over time.

## 5. Conclusions

Our cases highlight two confirmed congenital infections with SARS-CoV-2 in premature neonates. Abnormalities were noted on placental examination in the first case together with cultivatable virus and visualization of intracellular viral particles. The second case did not have pathological changes in the placenta which suggests that not all congenital infections will have placental pathological changes yet the Ct values in the placenta and cord blood were in the range of easily detectable and cultivatable virus. This report is unique based on the identification of SARS-CoV-2 congenital infection based on multiple modalities including molecular studies, histopathological analysis and viral culture. Further studies are required to understand the full spectrum of congenital clinical presentations, the effects of SARS-CoV-2 infection on placental tissues and the association between these changes and congenital infections.

## Figures and Tables

**Figure 1 viruses-15-01310-f001:**
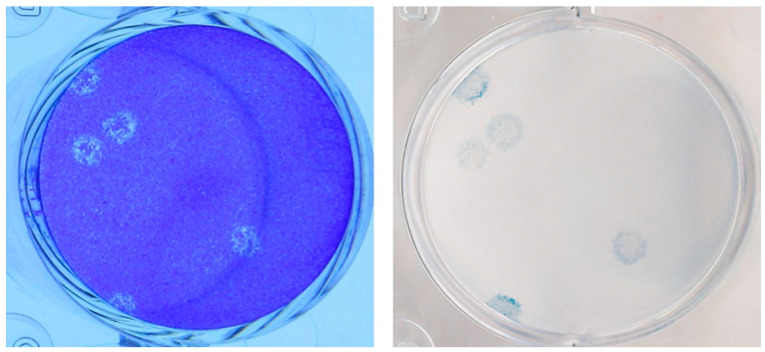
Placental tissue from Case 1-Three days post-infection the plates were fixed and stained with a crystal violet/formaldehyde solution to visualize the plaques (**left panel**). To confirm the identity of the virus seen in the left panel, the plates were destained with ethanol, immunostained with a 1:500 diluted rabbit anti-SARS-CoV-2 spike antibody (Prosci Inc., cat#3525) and visualized with a secondary goat anti-rabbit IgG (H+L) antibody coupled to horseradish peroxidase (Invitrogen, cat#G21234) and KPL TrueBlue peroxidase substrate (SeraCare, cat#5510-0052) (**right panel**). With both staining methods the plaques exhibited a halo-like appearance that is characteristic of SARS-CoV-2 cultured on Vero cells.

**Figure 2 viruses-15-01310-f002:**
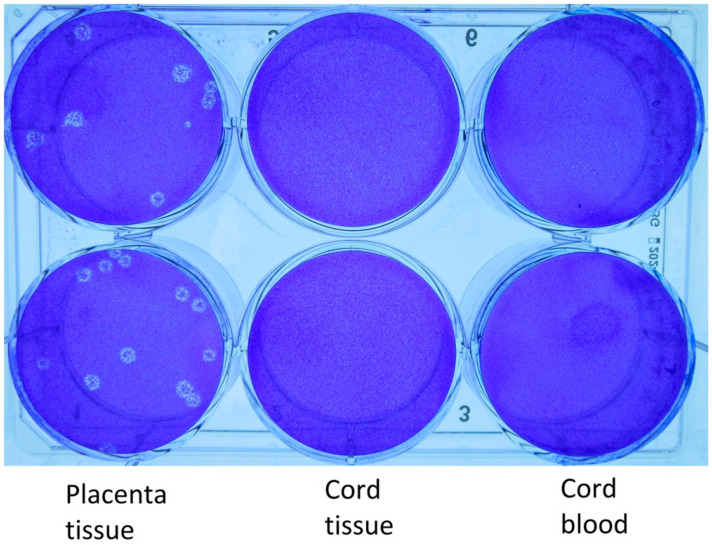
The cultures of placental tissue from Case 1 clearly reveal the plaque morphology and the confirmatory immunostaining for SARS-CoV-2. No growth of the virus was detected in cord tissue or cord blood.

**Figure 3 viruses-15-01310-f003:**
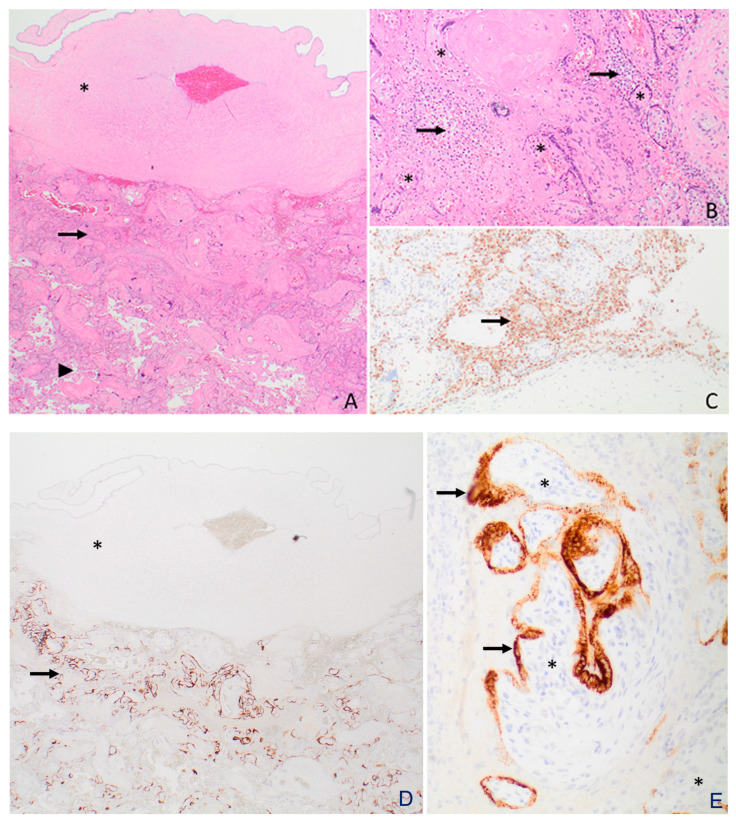
Placenta from Case 1. (**A**)The asterisk (*) indicates the chorionic plate (fetal surface) of the placenta. A histiocytic infiltrate (arrow) was present in the subchorionic (i.e., below the chorionic plate) intervillous space. Note that the intervillous space (arrowhead) below the subchorionic region (i.e., away from the fetal surface and closer to the maternal surface) did not contain a histiocytic infiltrate (H&E 20×). (**B**) Higher magnification shows numerous histiocytes (arrows) in the intervillous space. The asterisks indicate the chorionic villi (H&E, 100×). (**C**) The histiocytes (arrow) in the intervillous space were immunopositive for CD68, a histiocytic cell marker (CD68, 100×). (**D**) The arrow shows that SARS-CoV-2 spike protein immunopositivity was primarily localized to the subchorionic region (i.e., close to the fetal surface) of the placenta (SARS-CoV-2 spike protein, 20×). (**E**) Higher magnification shows that the SARS-CoV-2 spike protein immunostain was positive in the syncytiotrophoblasts (arrows) covering the chorionic villi (asterisks) (SARS-CoV-2 spike protein, 200×).

**Figure 4 viruses-15-01310-f004:**
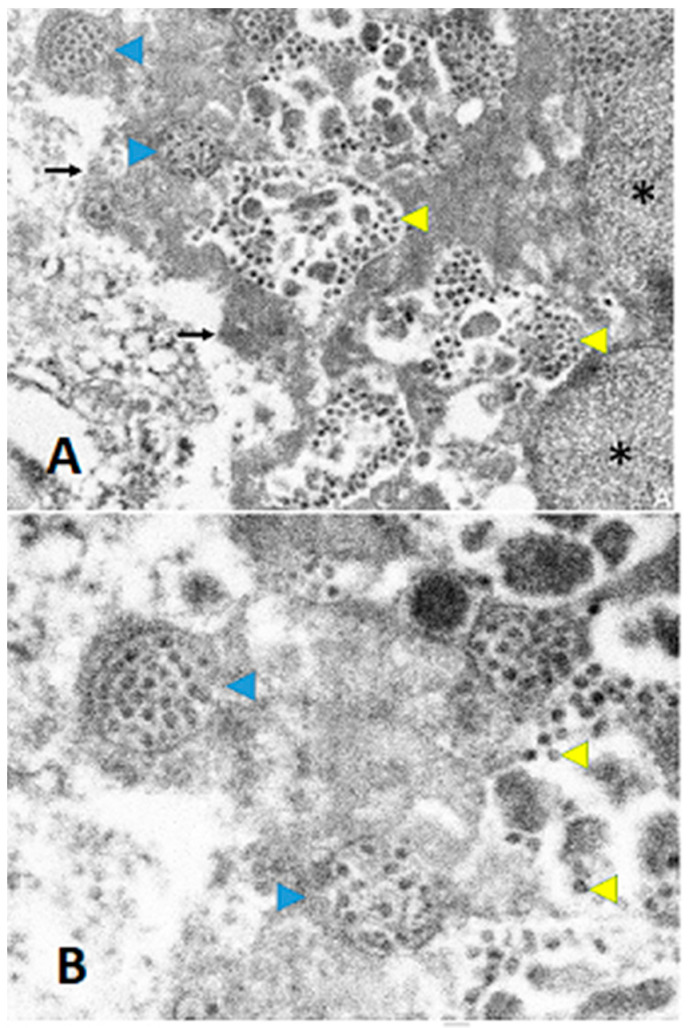
Electron microscopy of the placenta from Case 1. Some viral particles in the syncytiotrophoblasts were membrane-bound (blue arrowheads), whereas some viral particles in the syncytiotrophoblasts were not bound by any membranes (yellow arrowheads). The asterisks indicate the nuclei of the syncytiotrophoblasts, and the arrows indicate the apical surface of the syncytiotrophoblasts. (direct magnification is ×10,000 for (**A**), ×20,000 for (**B**)).

**Table 1 viruses-15-01310-t001:** Case 1: Summary of Samples Tested for SARS-CoV-2.

Specimen Type	RT-PCR Cycle Threshold (Ct) Value	Assay Target(s)	Viral Culture Titer (pfu/mL)	Estimated Viral Load (N Gene Copies/mL)	Whole Genome Sequencing Lineage
NP Swab Mother	21.0	E gene	ND	N/A	**B1.36.1**
NP Swab Neonate (24 h)	22.8/23.0	E gene	**4.0 × 10^2^**	N/A	**B1.36.1**
NP Swab Neonate (48 h)	17.5	E gene	N/A	N/A	Not performed
NP Swab Neonate (Day 14)	34.2/36.7	E gene	N/A	N/A	Not performed
NP Swab Neonate (Day 21)	29.8/31.4	E gene	N/A	N/A	Not performed
NP Swab Neonate (Day 31)	NEGATIVE	E gene	N/A	N/A	N/A
CSF (Day 7)	NEGATIVE	E gene	N/A	N/A	N/A
CordBlood ** (1:1 UTM)	33.3	N gene	ND	**7.0 × 10^2^**	Not performed
Cord blood	37.2	E gene	ND	N/A	Not performed
Placenta	24.9	E gene	N/A	N/A	**B1.36.1**
Placenta *	16.0	N gene	**2.8 × 10^2^**	**8.9 × 10^7^**	Not performed
Cord tissue *	30.2	N gene	ND	**5.9 × 10^3^**	Not performed
Serology	Specimen	Target	Result	N/A	Manufacturers
Blood	Plasma (Day 6 of life)	COVID-19 Antibody IgG (nucleocapsid & spike)	NEGATIVE		Abbott Laboratories & DiaSorin

** RNA was extracted from 140 µL of sample. * RNA was extracted from 28 µL of sample; NP—nasopharyngeal, ND—not detected, N/A—not applicable; E gene Ct values determined by the Alberta ProvLab assay [16]. N gene by the US CDC assay [18].

**Table 2 viruses-15-01310-t002:** Case 2: Summary of Samples Tested for SARS-CoV-2.

RT-PCR Specimen Type	RT-PCR Cycle Threshold (Ct) Value	Assay Target(s)	Assay
Mother NP swab	23.5/25.4	S gene/ORF1ab	Simplexa^®^
Placenta	19.5	E gene	ProvLab E gene
Cord Blood	24.9	E gene	ProvLab E gene
Neonate NP Swab–24 h	26.4/29.1	S gene/ORF1ab	Simplexa^®^
Neonate NP Swab–48 h	14/15	S gene/ORF1ab	Simplexa^®^
Neonate NP Swab–Day 18	26.5/27	S gene/ORF1ab	Simplexa^®^
Serology	Result	Assay Target	Assay
Neonate Blood–Plasma	NEGATIVE (Day 1 of life)	COVID-19 Antibody IgG (Nucleocapsid & Spike)	Abbott Laboratories (Nucleocapsid) & DiaSorin (Spike)

## Data Availability

Additional data supporting the reported results can be obtained by contacting the authors.

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
