# Peer review of "Congenital SARS-CoV-2 Infection in Two Neonates with Confirmation by Viral Culture of the Placenta in One Case"

_viruses, 2023, doi:10.3390/v15061310_

Round 1

Reviewer 1 Report

The paper is interesting and I think that the quality of presentation and the scientific soundness are high. I have only few suggestions for the authors that might improve the presentation of their study:

-in the legend of Figure 1 I think it should be indicated that the viral cultures were obtained from the specimens from case 1 and also from which samples (placenta, I guess).

-in the section 3.3 lines 198 and 199 the authors state: "the placenta of case 1 showed that the SARS-CoV-2 spike protein immunochemical stain (Figure 3) was positive in a subset of trophoblasts in the subchorionic region". In the Figure 3 I see only an immunohistochemical stain for CD68 (panel C) as the legend explains, and no immunohistochemical stain for the SARS-CoV-2 spike protein.

-in the same section 3.3 line 202 and 203  the authors state that in case 2 there was no evidence of inflammation in the placental tissue examined and I think that this should be shown in a picture in comparison to case 1.

-in the supplementary materials section the authors provide a table with a summary of case 1 laboratory investigations. In the table there are two examinations of CSF, on day 5 and on day 7, but RT-PCR for SARS-CoV-2 was performed as shown in table 1 line 112 on day 7 only, resulting negative. It has been performed on day 5? On day 6 a cranial ultrasound showed a right-sided germinal matrix hemorrhage. What was a possible cause of this hemorrhage, sepsis or SARS-CoV-2 infection? There was a further cranial ultrasound to check the hemorrhage evolution?

-The authors checked several specimens for SARS-CoV-2 by RT-PCR (NP swab, CSF, cord blood, placenta, cord tissue). Did they check for SARS-CoV-2 in the mother's blood and in the blood of case 1?

-Both mothers of case 1 and case 2 had a mild COVID-19 infection, but case 1 infant had a worse outcome (even if he improved and was discharged in good conditions on day 54). I wonder if the two mothers were both vaccinated for COVID and had a similar response to the vaccination (have they been checked for anti-spike IgG and neutralization titer?) How can the authors explain the fact that both case 1 and case2 had no COVID-19 antibody IgG? 

Reviewer 2 Report

Dear Editor,

Thank you for asking me to review the work of Vayalumkal et al. These observations are a reminder that congenital infections are still possible without causing severe complications for the newborn. The manuscript is generally well written (with the exception of the discussion which is disorganised). The reading of this manuscript raises several questions. The first of these concerns the willingness to publish case number 2, which suffers from several important flaws

- The authors insist on the added value of viral culture, whereas they did not perform a viral culture for case number 2.

- The clinical, biological and histopathological description of case number 2 is much less detailed.

- The authors describe an absence of placental lesions in case 2, whereas vertical transmission through the placenta appears likely. This raises a major question when we know that there is a very strong association between placental infection and placental lesions (it is surprising that this article is not cited and discussed: DOI: 10.5858/arpa.2020-0771-SA )

In general, please cite peer-reviewed scientific literature rather than pre-print studies.

Regarding case 1, premature rupture of membranes and vaginal delivery make vertical transmission "per partum" a possibility and this is not discussed. The authors state that SARS-CoV-2 cannot be excreted vaginally and this is not true (see below). The authors state that this is a confirmed transplacental infection but this must be discussed.

Could the authors provide the month and year of infection to put it in the context of the different pandemic waves?

Please also respond to my comments below:

Line 41: ref 5 useless

Line 46: please replace ref 11 with an original publication and not a meta-analysis

Line 53: please cite a reviewed article rather than a pre-print (reference 14)

Materials and Methods: please reorganise the section (2.1; 2.2; etc) by putting in the last sub-section the information about the ethics committee, consent and anonymisation of data

Please give details of the techniques used for histological placental analysis (including immunohistochemical staining).

Please quote now the WHO classification of vertical transmission of SARS-CoV-2 and refer to it precisely (confirmed or likely confirmed) about your cases and use this nomenclature throughout the manuscript.

Line 91: Was this the first pregnancy? Please specify that this was a singleton pregnancy.

Line 98: Please provide the results of the laboratory tests performed on arrival (C-reactive protein, APTT, transaminitis, lymphocyte count, D-dimers, platelet count).

Line 98: What was the cervical length measurement on admission? Was the patient describing uterine contractions?

Line 99: Please provide a copy of the cardiotocograph and classify it according to the ACOG classification (https://www.mnhospitals.org/Portals/0/Documents/patientsafety/Perinatal/3a_ACOG%20Bulletin%20106.pdf)

Line 101: please quote the WHO classification for mild maternal infection.

Table 1: Could the authors provide an estimate of the viral load in each of the compartments tested by RT-PCR

Line 128: did the newborn receive a brain image (MRI) because of the reported neurological symptoms?

Line 137: typo "SARS-CoV-2”

Line 149: same remarks: please provide a copy of the cardiotocography and classify it according to the ACOG classification. Please provide the results of the biological tests performed on the patient's arrival (C-reactive protein, APTT, transaminitis, lymphocyte count, D-dimers, platelet count).

Table 2: Could the authors give an estimate of the viral load of each of the compartments tested by RT-PCR. Same remark: did the newborn receive brain imaging? If so, what were the results?

Could the authors provide the cord pH (+ arterial lactates) for both cases.

What was the vaccination status of both patients?

Figure 2: How do you explain the negativity of the viral culture for the cord blood when the RT-PCR is positive (table 1) for the same sampling site?

Line 199: You state that there is positive immunohistochemical labelling of the SARS-CoV-2 spike protein in Figure 3 but it is not shown in Figure 3.

Figure 4 is not referred to in the text.

Line 226: this assertion is false "Perinatal acquisition re-226 lated to prolonged rupture of membranes is considered unlikely as previous studies have 227 not shown vaginal fluids or the female genital tract of pregnant women to contain the 228 virus"

Please refer to  DOI: 10.1016/j.jogoh.2023.102547

Discussion :

The discussion is not well organised. I think it is important to reorder the main ideas so as not to repeat yourself too much.

I suggest the following plan:

- Principal Findings

- Results in the Context of What is Known

- Clinical Implications

- Research Implications

- Strengths and Limitations

You may wish to cite this manuscript (DOI: 10.1038/s41467-020-18982-9)

Line 305: "Our report highlights the importance of viral cultures in assessing the potential for transplacental transmission.” In addition, I would have liked the authors to highlight the added value of viral culture compared to the techniques already available for qualification as congenital infection according to WHO classification.

Supplementary materials : cranial ultrasound is not part of laboratory investigations (table S1 title)

Please also provide the biological parameters for case 2

Round 2

Reviewer 2 Report

Dear Editor,

the authors have taken into account the majority of my comments and suggestions and the manuscript has been improved.

However, I still do not understand the wish to maintain case number 2 which has not been tested by viral culture when it is precisely the title of the article. Moreover, case number 2 has much less detail as I already mentioned in my first proofreading.

Furthermore, I do not understand why the authors cannot provide a copy of the CTG recording for both cases when they were able to review it to correctly classify the fetal heart rate abnormalities according to the ACOG classification.

I think it is important to focus on case number 1. I suggest that the authors publish this article as a case report by removing the data of case number 2 which does not bring any added value and even has the opposite effect because it does not fit in the scope of the title of the article.

I think this work deserves to be published if the authors agree to focus on the description of case number 1.

Round 3

Reviewer 2 Report

The authors decided not to take into account my recommendations concerning case number 2, which does not provide any relevant and new information and is not very detailed.
Furthermore, I had requested that the CTG analysis be made available, at least as an appendix: the authors also decided to go against my recommendation.
In the end, the authors did not make a revision according to my recommendations. My opinion remains the same and I cannot approve the publication as it stands.
